# Delayed Response and Random Backoff First for Low-Power Random Access of IoT Devices with Poor Channel Conditions

**DOI:** 10.3390/s23239556

**Published:** 2023-12-01

**Authors:** Minjoong Rim

**Affiliations:** Department of Information and Communication Engineering, Dongguk University, Seoul 04620, Republic of Korea; minjoong@dongguk.edu; Tel.: +82-2-2260-3595

**Keywords:** random access, IoT, low power, repetition, response, random backoff, interference cancellation

## Abstract

As IoT services become more active, the density of IoT devices is increasing, and massive connectivity technology is needed to support numerous devices simultaneously. In addition, IoT devices are often battery-powered, and during random access, it is necessary to reduce the power consumption to extend the lifetime of the devices. In particular, devices with poor channels need to send at a very low transmission rate through a large number of repetitions, and longer packet lengths can increase the probability of collisions, increasing the power consumption while shortening the lifetime of the IoT system. Dividing devices into groups based on the number of repetitions and allocating different resources to each group can reduce collisions for bad-channel devices, but it can be difficult to support large connections, due to the inefficient use of resources. This paper proposes schemes to reduce the collision probability of bad-channel devices while allowing IoT devices to use shared resources, instead of dividing resources by groups. There are two versions of the proposed schemes. The first method reduces collisions by delaying the response of a bad-channel device, and in the meantime, eliminating interference from other devices, assuming that the bad-channel device is not sensitive to delay. Instead of checking the response, and then, performing a random backoff when no acknowledgement packet is received, the second proposed method reverses the order of response checking and random backoff, that is, it first performs a random backoff, and then, checks the response to decide whether to retransmit. The proposed method can increase the lifetime of the IoT system by reducing the collision probability of a bad-channel device, without degrading the performance of other devices.

## 1. Introduction

In an Internet of Things (IoT) system, devices transmit data to the network through random access. As IoT services flourish, the density of IoT devices is increasing, requiring massive connectivity technologies that can effectively support simultaneous random access by numerous devices [1,2,3,4,5,6,7,8,9,10,11,12]. In addition, since IoT devices are often battery-powered, it is necessary to reduce the power consumption as much as possible to extend the lifetime of the devices when transmitting through random access [13,14,15,16,17,18,19]. Especially when the number of devices attempting random access is large, the probability of retransmission may increase due to a large number of collisions, which can also increase the power consumption of the devices. For devices with low residual energy or high power consumption, it is necessary to reduce the retransmission probability to suppress the power consumption.

Some devices transmitting to a single base station may have good channel conditions, while others may have poor channel conditions due to obstacles or a long distance from the base station. In particular, devices located deep indoors or underground may experience very poor channel conditions. Devices with good channel conditions can transmit using high modulation and coding schemes and low transmission power, while devices with poor channel conditions may need to transmit at very low rates through a large number of repetitions and at the maximum transmission power to get a packet to the base station [19,20,21,22,23]. Particularly for devices with very poor channel conditions, the number of repetitions can become very large to obtain sufficient receiving energy at the base station. Assuming that slotted Aloha is used for random access, a device with good channel conditions may be able to transmit a packet in a single slot, while a device with poor channel conditions may need to transmit a packet over a large number of slots. 

If a packet is transmitted at full transmission power for a long time, the transmission energy increases, which can reduce the lifetime of the device [18,19]. A longer packet transmission time can also increase the collision probability and the number of retransmissions, which can further shorten the lifetime of the device. If different devices use different numbers of slots to transmit a packet, a packet that occupies a larger number of slots has less chance of being successfully received at the base station, due to collisions in one or more of the slots. As the packet length increases, the probability of collision significantly increases, and the power consumption may increase due to a large number of retransmissions, especially when the number of devices is large. 

To reduce collisions on devices with poor channel conditions, devices can be grouped according to their channel conditions, and dedicated random access resources can be allocated to each group [19,20,21,22,23], or algorithms that make efficient use of given resources can be applied, such as the multi-armed bandit algorithm [24,25]. Ensuring sufficient resources for bad-channel devices can reduce the collision probability of the bad-channel devices, but it may not be easy to use resources efficiently, making it difficult to support numerous devices. When groups are divided based on the number of repetitions, dividing devices into a large number of groups may make it difficult to use each resource efficiently, while dividing them into a small number of groups may prevent the use of the appropriate number of repetitions to support a large number of devices. In addition, if separate resources are allocated based on the number of repetitions, it may not be easy to determine the maximum number of groups, because the worst-case channel condition may be known in advance. Rather than using separate resources between groups depending on channel conditions or packet lengths, this paper considers methods to support multiple groups of devices using interference cancellation techniques on a non-separated resource. 

Even when packets from multiple devices collide, all or multiple packets can be received using non-orthogonal multiple access (NOMA) with interference cancellation [26,27,28,29,30,31,32,33,34,35,36,37]. NOMA techniques first decode packets from devices with good channels; then, they remove the interference from those packets, and attempt to decode devices with bad channels. However, NOMA may need to sufficiently increase the received power of good-channel devices relative to that of bad-channel devices or adjust the packet length of good-channel devices to that of the worst-channel device, which may increase the power consumption of good-channel devices. In real systems, interference cancellation may not be perfect due to inaccuracies in channel estimation, frequency offset, in-band emission, etc., while unnecessarily boosting the power of packets from good-channel devices by increasing the receiving power or the number of repetitions may also increase the residual interference remaining after cancellation, making it difficult to decode bad-channel devices [38,39,40]. NOMA is a technique that generates a large amount of interference, and performance may be degraded in conjunction with other techniques that use interference cancellation [41,42]. 

Another technique for eliminating interference in the presence of collisions is coded random access [43,44,45,46,47,48]. In coded random access, a device transmits multiple packets. If one of the transmitted packets is successfully received without a collision, the interference from the other packets is eliminated to resolve the collisions. Coded random access can be used in conjunction with other techniques that use interference cancellation, but it can increase the power consumption because a device transmits multiple packets at once. 

To increase the lifetime of an IoT system while supporting a large number of IoT devices, a method is needed to reduce the collision probability of devices with poor channel conditions without increasing the transmission power or collision probability of other devices. Instead of using NOMA technology or coded random access, this paper applies the technique described in [49] to perform interference cancellation using retransmitted packets. The contributions of this paper are as follows: (1)In this paper, instead of using separate resources by group, multiple groups use shared resource to efficiently use a random access resource. Using the proposed techniques, even with the shared resource, the collision probability of bad-channel devices can be reduced without increasing the collision probability of other devices.(2)The proposed methods perform interference cancellation for bad-channel devices, but do not increase the transmission power of other devices. The interference cancellation is performed at the base station and does not sacrifice the transmission power of good-channel devices.(3)There are two versions of the proposed schemes. The first method, called Delayed Response, delays the response of a packet to eliminate the interference to the packet in the meantime. The second method, called Random Backoff First, reverses the order of response checking and random backoff. A system can choose the appropriate method based on the system requirements or use a combination of both methods.(4)The proposed method does not sacrifice the performance of good-channel devices for the sake of bad-channel devices. The interference cancellation significantly reduces the collision probability of bad-channel devices while also reducing the collision probability of good-channel devices to some extent.

This paper is organized as follows. Section 2 briefly describes the system model, while Section 3 discusses the two proposed methods for reducing the collision probability of bad-channel devices. Section 4 discusses interference cancellation using retransmitted packets, and then, Section 5 shows the feasibility of the proposed methods through the simulation results. Finally, Section 6 presents the conclusions. 

## 2. System Model 

Figure 1 shows the IoT system considered in this paper, where $N^{device}$ devices perform uplink transmission to a base station using slotted Aloha. The set of $N^{device}$ devices is called ${Set}^{device}$. 

Suppose a packet from device i occupies $L_{i}^{TX}$ slots without repetition and occupies $R_{i}L_{i}^{TX}$ slots with $R_{i}$ repetitions. The reception failure probability of device i, denoted as $F_{i}$, can be written as:(1)$$F_{i}={1-(1-F}_{i}^{channel}{)(1-F}_{i}^{collision})$$
where $F_{i}^{channel}$ is the reception failure probability of device i when there is no collision, and $F_{i}^{collision}$ is the collision probability when device i is transmitting. If the signal-to-noise-plus-interference ratio (SINR) of device i without repetition is $\gamma_{i}$, $F_{i}^{channel}$ can be written as the probability that the received SINR falls below a certain threshold, $\Gamma_{i}$:(2)$$F_{i}^{channel}=Ρ(R_{i}\;\gamma_{i}<\Gamma_{i})$$

The above equation shows that increasing the number of repetitions reduces the probability of reception failure. 

To simplify the consideration of the collision probability, it is assumed that each device performs a slotted Aloha transmission with $R_{i}{L\;}_{i}^{TX}$ slots, which is a power of 2. Assuming that, on average, one packet must be successfully transmitted every $P_{i}$ slot period, on average, one packet must be transmitted every ${\left(1-F_{i}\right)P}_{i}$ slot period. The collision probability with device j when device i is transmitting can be written as:(3)$$F_{i,\;j}^{collision}=\frac{\max\left(R_{i}L_{i}^{TX},{\;R_{j}L}_{j}^{TX}\right)}{{(1-F}_{j})P_{j}}$$

Therefore, the collision probability when device i transmits can be expressed as follows:(4)$$F_{i}^{collision}=1-\prod_{j\in {Set}_{}^{device},j\neq i}\left(1-F_{i,\;j}^{collision}\right)$$

Figure 2 shows that devices with long packet lengths due to a large number of repetitions can have a high probability of reception failure. Devices with poor channel conditions consume a lot of transmission energy, so it may be necessary to reduce the collision probability. In the case of a reception failure, it may not be known which device has transmitted, so the base station does not send a response packet. It can be assumed that a response packet will be sent as an acknowledgement (ACK) packet only if the reception is successful. 

For devices with poor channel conditions, the energy consumption per transmission is high, and a higher collision probability can shorten the lifetime of the devices. Allocating resources based on packet lengths can increase the lifetime of devices with poor channel conditions, but if resources are not distributed carefully, a large number of devices may not be accommodated. 

Suppose an ACK packet from the base station to device i occupies $L_{i}^{RX}$ slots. If the transmission energy per slot is $E_{i}^{TX}$, the receiving energy per slot is $E_{i}^{RX}$, and the idle energy per slot is $E_{i}^{idle}$, then the energy consumed per $P_{i}$ slot cycles for device i can be written as follows:(5)$$E_{i}^{period}=\frac{{R_{i}\;L}_{i}^{TX}E_{i}^{TX}+{L_{i}^{RX}E}_{i}^{RX}}{{1-F}_{i}}+\left(P_{i}-\frac{{R_{i}\;L}_{i}^{TX}+L_{i}^{RX}}{{1-F}_{i}}\right)E_{i}^{idle}$$

If the remaining energy of device i is called $E_{i}^{remained}$, the remaining survival time can be written as follows:(6)$$T_{i}=\frac{P_{i}{\;E}_{i}^{remained}}{E_{i}^{period}}$$

For devices with a low remaining lifetime, it may be necessary to make efforts to increase the lifetime. Assuming that the transmission power is very large compared to the receiving power or the idle power, Equation (5) can be approximated as follows:(7)$$E_{i}^{period}\approx \frac{{R_{i}\;L}_{i}^{TX}E_{i}^{TX}}{\left(1-F_{i}^{channel}\right)\left(1-F_{i}^{collision}\right)}$$

Therefore, Equation (6) can be rewritten as follows: (8)$$T_{i}\approx \frac{{\left(1-F_{i}^{channel}\right)\left(1-F_{i}^{collision}\right)\;P}_{i}{\;E}_{i}^{remained}}{{R_{i}\;L}_{i}^{TX}E_{i}^{TX}}$$

From the above equation, as the number of repetitions increases due to poor channel conditions, the remaining lifetime decreases. Therefore, it may be necessary to increase the lifetime of the system by reducing the collision probability of devices with many repetitions. 

## 3. Proposed Random Access Schemes

### 3.1. Delayed Response

This paper proposes two methods by applying the interference cancellation technique described in [49] to eliminate interference for devices with bad channel conditions. The first method, called Delayed Response, as shown in Figure 3 and Figure 4, postpones the response for a device with a bad channel condition under the assumption that the data transmission is not delay-sensitive, and in the meantime, reduces collisions by eliminating interference from other devices. The base station has a buffer that can store the received signal during the maximum response time, and it attempts to detect and decode the signal just before each device’s response time. 

Packets from other devices that collide with a bad-channel device can be retransmitted before the base station sends the Delayed Response for the bad-channel device. If the retransmission of packets from other devices is successful, the base station finds the slot positions of previously transmitted packets to recover the packet signals and subtracts them from the received signal to eliminate interference. When the response delay elapses, the base station determines whether the reception is successful by detecting and decoding the signal, and sends a response if the decoding is successful. Increasing the response delay improves interference cancellation performance but may introduce unnecessary transmission latency. 

When the device set ${Set}^{device}$ is divided into $N_{}^{group}$ groups according to the Delayed Response time, ${Set}_{1}^{device}$ is the group of devices with no delay, and ${Set}_{n}^{device}$ is the group of devices with a response delay greater than that of ${Set}_{n-1}^{device}$. The groups can be divided according to the number of repetitions or the remaining lifetime. Assume that collisions caused by devices from ${Set}_{1}^{device}$ to ${Set}_{n-1}^{device}$ can mostly be eliminated by interference cancellation during the delay time ${Set}_{n}^{device}$. If the group to which device i belongs is called ${Set}_{group(i)}^{device}$, the collision probability of device i can be expressed as:(9)$$F_{i}^{collision,new}\approx 1-\prod_{n=group(i)}^{N_{}^{group}}\prod_{j\in {Set}_{n}^{device},\;j\neq i}^{}(1-F_{i,\;j}^{collision})$$

In this case, Equation (8) can be rewritten as follows:(10)$$T_{i}^{new}\approx \frac{{\left(1-F_{i}^{channel}\right)\left(1-F_{i}^{collision,new}\right)\;P}_{i}{\;E}_{i}^{remained}}{{R_{i}\;L}_{i}^{TX}E_{i}^{TX}}$$

If $F_{i}^{collision}$ was originally very large and the number of devices with the same or a greater response delay compared to device i is small, so that $F_{i}^{collision,\;new}$ can be greatly reduced, the remaining lifetime can be significantly improved. In particular, when device i belongs to ${Set}_{N^{delay}}^{device}$ and the number of devices in ${Set}_{N^{delay}}^{device}$ is very small, Equation (9) becomes $F_{i}^{collision,new}\approx 0$ and Equation (1) can be approximated as $F_{i}^{new}\approx F_{i}^{channel}$; therefore, Equation (10) can be rewritten as follows:(11)$$T_{i}^{new}\approx \frac{{{(1-F}_{i}^{channel})\;P}_{i}{\;E}_{i}^{remained}}{{R_{i}\;L}_{i}^{TX}E_{i}^{TX}}$$

Delaying a response can cause a transmission delay for the device and can only be applied to delay-tolerant services. Delaying a response does not harm other devices. Rather, it may also reduce the collision probability of other devices by not participating in retransmissions. 

### 3.2. Random Backoff First

The method proposed in the previous subsection may increase transmission latency due to the response delay. To eliminate interference using retransmitted packets, the transmission latency cannot be reduced to a small value. However, suppressing unnecessary response delay can be considered. In general, after checking the response to a transmitted packet, if the reception fails, the device performs a random backoff by selecting a random number from the contention window to reduce the collision probability of the retransmitted packet. In the second proposed method, called Random Backoff First, as shown in Figure 5 and Figure 6, instead of checking the response, and then, performing a random backoff for retransmission when the reception fails, the device first performs a random backoff, checks the response, and then, performs retransmission when the reception fails. 

To support this, a random number selected by a device from the contention window must be determined via a hash function so that the base station knows the backoff value using the same hash function as the device. The base station uses the data and the location of a received packet to obtain the location information of the previously transmitted packets; for example, the base station extracts the device ID, the packet sequence number, etc. from the received packet, and the number of retransmissions using the slot of the received packet, which are used to find the slots of the previously transmitted packets [49]. The backoff value is generated by a hash function and appears to other devices as a random number, but the base station can find the value used by the device. 

As shown in Figure 7 and Figure 8, the IoT base station attempts to decode the packet each time interference cancellation is performed. If a packet is successfully decoded, the backoff time can be found using the hash function, and if it has not yet elapsed, as shown in Figure 7, the ACK packet is transmitted at the appropriate time slot. If a packet can be successfully decoded after some time, but the backoff time has elapsed, as shown in Figure 8, no ACK packet is sent, and the reception is considered to have failed. If the number chosen by the hash function for random backoff is a small value, as shown in Figure 8, interference cancellation will not be sufficient, and the packet may be retransmitted with a higher probability. On the other hand, if the number selected by the hash function is large, as shown in Figure 7, sufficient interference cancellation can be performed, and the reception is likely to be successful. Table 1 summarizes the differences between the Delayed Response and Random Backoff First methods.

The Random Backoff First method has the advantage of no additional response delay, but if the backoff value is small, it may not provide sufficient interference cancellation. In this case, it may be necessary to use a combination of the Delayed Response and Random Backoff First methods. The Random Backoff First method may increase the load on the base station compared to the Delayed Response method, because it requires multiple decoding attempts as interference cancellation progresses until decoding is successful.

## 4. Interference Cancellation

This section describes how to perform interference cancellation using retransmitted packets by applying the technique described in [49]. An attempt is made to eliminate interference caused by short packets, and interference caused by long packets may not be considered for interference cancellation. Consider device i transmitting with slotted Aloha in units of packet length $L\equiv R_{i}L_{i}^{TX}$. The slot number of each slot is called Slot, and Slot is considered as a combination of the virtual superframe Frame in units of M slots and slot position Pos within the virtual superframe:(12)Frame=Slot div M
(13)Pos=Slot mod M

In the above equation, div indicates the quotient, and mod indicates the remainder. 

In this paper, to reduce the collision when a device is about to transmit a new packet, the slot position within the superframe is randomly determined using the following hash function. To reduce unnecessary complexity, the device index i is omitted in the following equations. The virtual superframe to be transmitted by a device is called ${Frame}_{0}$, ID is the device ID, and SN is the packet sequence number. The slot position in the superframe is determined as follows:(14)$${Pos}_{0}=H_{slot}\left(ID,SN,{Frame}_{0},L\right)$$

The slot position within a superframe determined in this way will have different values for each device, each packet sequence number, and each superframe, so it will appear random to other devices. The transmission slot is determined as follows:(15)$${Slot}_{0}={Frame}_{0}\times M+{Pos}_{0}$$

When retransmission is performed due to a transmission failure, let $d_{k}$ be the superframe interval between the k-th transmission and the (k−1)-th transmission (the 0-th transmission is considered to be the initial transmission). $d_{k}$ is determined using a hash function between the minimum interval $D_{min}$ and the maximum interval $D_{min}+{CW}_{k}-1$, where ${CW}_{k}$ is the contention window of the k-th retransmission:(16)$${Frame}_{k}={Frame}_{k-1}+d_{k}$$
(17)$$d_{k}=H_{backoff}\left(ID,SN,k,L\right)$$

The frame interval has different values for each device, each packet sequence number, and each number of retransmissions, so it appears random to other devices. Once the superframe for the k-th retransmission is determined, if k is less than M, the slot position in the superframe is determined as follows:(18)$${Pos}_{k}=\left(H_{slot}\left(ID,SN,{Frame}_{k},L\right)+k\right)\;mod\;M$$

If k is equal to or greater than M, then no further interference cancellation is performed, and the slot position is determined as follows:(19)$${Pos}_{k}=H_{slot}\left(ID,SN,{Frame}_{k},L\right)$$
(20)$${Slot}_{k}={Frame}_{k}\times M+{Pos}_{k}$$

The slot position of a packet transmitted via this method appears random to other devices, but when a packet is received at the base station, if it is a retransmitted packet, the slot positions of the previously transmitted packets can be found. Suppose a packet is received in a certain slot Slot, and its virtual superframe and the slot position in the superframe can be found using Equations (12) and (13). If the number of retransmissions k is less than M, it can be found as follows: (21)$$k=\left(Pos-H_{slot}\left(ID,\;SN,\;Frame,\;L\right)+M\right)\;mod\;M$$

If k found using the above equation is zero, it is either an initial transmission or the number of retransmissions is equal to or greater than M, and interference cancellation cannot be performed. If k is not zero, it is a retransmitted packet with the number of retransmissions k, and the slot position of the previously transmitted packet can be found as follows: (22)$${Frame}_{k-1}=Frame-H_{backoff}\left(ID,SN,k,L\right)$$
(23)$${Pos}_{k-1}=\left(H_{slot}\left(ID,SN,{Frame}_{k-1},L\right)+k-1\right)\;mod\;M$$
(24)$${Slot}_{k-1}={Frame}_{k-1}\times M+{Pos}_{k-1}$$

After the previously transmitted packet is recovered and removed from the received signal, signal detection is performed to find another packet. In the same way, all slot positions of previous transmissions can be found, and interference cancellation and signal detection can be performed. The slot position of the (k−i)-th retransmission can be found as follows.
(25)$${Frame}_{k-i}=Frame-\sum_{j=0}^{i}H_{backoff}\left(ID,SN,k-j,L\right)$$
(26)$${Pos}_{k-i}=\left(H_{slot}\left(ID,SN,{Frame}_{k-i},L\right)+k-1\right)\;mod\;M$$
(27)$${Slot}_{k-i}={Frame}_{k-i}\times M+{Pos}_{k-i}$$

In this way, interference caused by short packets can be eliminated without a separate message or signal. Signal detection for devices with poor channel conditions is performed after all interference cancellation is complete, and the ACK packet is sent if the decoding is successful.

## 5. Simulation Results

### 5.1. When a Good-Channel Device Does Not Perform Random Backoff

In the simulation, there are $N^{device}$ devices performing random access to a single base station, and each device transmits one packet every 512 slots with a random start time. It is assumed that only one of the $N^{device}$ devices has a bad channel, while the other $N^{device}-1$ devices have good channels. A packet occupies one slot for the good-channel devices, and the virtual superframe size M is 8. The cancellation of interference caused by a good-channel device can be performed for up to seven previously transmitted packets, and interference caused by a bad-channel device is not canceled. The detailed simulation parameters are shown in Table 2. 

Delays for decoding or sending ACK packets are not considered in the simulation, but good-channel devices are assumed not to transmit immediately in the next superframe when retransmitting, considering the time to decode the packet, send the ACK packet, decode the ACK packet, etc., but an interval of one superframe is added between retransmission frames. The collision probability of the bad-channel device is measured by varying $N^{device}$.

Consider a case where good-channel devices do not perform random backoff in units of the superframe, so that the number of previously transmitted packets for interference cancellation is fixed according to the response delay of the bad-channel device. Even without random backoff in units of the superframe, good-channel devices can reduce collisions of retransmitted packets by randomly selecting slot positions in the virtual superframe. 

Figure 9 shows the collision probability of the bad-channel device without interference cancellation as a function of the number of devices, as the packet length of the bad-channel device increases from 1 to 16. With a packet length of 16, a single packet occupies two superframes. Interference caused by the bad-channel device is not canceled, and a packet from the bad-channel device can occupy resources beyond the superframe. As the packet length of the bad-channel device increases, the collision probability increases dramatically, and the number of retransmissions increases, which can shorten the lifetime of the bad-channel device. As the density of IoT devices increases and the number of devices performing random access to a single base station increases, the collision probability of a bad-channel device can become considerable, especially when the packet length of the bad-channel device is long. Even in an environment where a device with a short packet length may not have a significant collision probability, the collision probability of the bad-channel device can be very large due to the long packet length. 

Figure 10 shows the collision probability of a bad-channel device upon varying the response delay of the bad-channel device. A packet from the bad-channel device occupies 16 slots, or two superframes, and delay 1 in the figure means a delay of 16 slots. When reception fails, a good-channel device retransmits every two superframes without random backoff, so the delay in the figure represents the number of interference cancellations, unless the delay is greater than 7. Since interference cancellation is performed on up to seven previously transmitted packets, a delay of 8 indicates maximum interference cancellation, in other words, cancellation of up to seven previously transmitted packets. By increasing the response delay and the number of packets considered for interference cancellation, the collision probability of the bad-channel device can be significantly reduced, and the number of retransmissions can be reduced. The figure shows that even a small increase in the response delay can significantly improve performance, so there is no need to delay the response excessively. 

Figure 11 shows the collision probability when, instead of delaying the response, a random backoff is performed first, and then, the response is checked to determine whether to retransmit. By varying the size of the contention window until good-channel devices retransmit 0, (0~1), (0~3), (0~7), and (0~15) times, the collision probability of the bad-channel device is examined. A packet from the bad-channel device occupies 16 slots. If a large random number is selected from the contention window when performing random backoff, the interference from other devices can be largely eliminated and the performance can be improved to some extent, even without a response delay. However, compared to Figure 10, there is a difficulty in reducing the collision probability. 

Figure 12 shows the collision probability when using both Delayed Response and Random Backoff First. To ensure the minimum response delay, a minimum interval between packets is included when using random backoff. The collision probability of the bad-channel device is examined by varying the size of the contention window until good-channel devices retransmit 1, (1~2), (1~4), (1~8), and (1~16) times. One packet from the bad-channel device occupies 16 slots. Compared to Figure 11, the performance is greatly improved by increasing the minimum interval between retransmission packets, so that a good-channel device with a collision can retransmit at least once before sending the response to the bad-channel device. 

Figure 13 and Figure 14 show how interference cancellation affects good-channel devices. In this simulation, there is no bad-channel device, and all devices have good channel conditions, i.e., a packet always occupies one slot. There is no response delay for good-channel devices, so they do not receive direct benefits like the bad-channel device in the previous simulations. However, the interference from colliding packets can be removed and decoding is sometimes successful, so there is a chance that the performance will improve. Figure 13 shows the collision probability of a good-channel device when eliminating up to seven previously transmitted packets. Since good-channel devices have a relatively low collision probability, a wider range of device counts (1~150 devices) is considered. When the collision probability is low due to a small number of devices, interference cancellation has little effect, but as the number of devices increases, and thus, the collision probability increases, interference cancellation can meaningfully reduce the collision probability of good-channel devices. In addition to significantly reducing the collision probability of bad-channel devices with Delayed Response, interference cancellation can also improve the performance of good-channel devices without Delayed Response. Figure 14 shows the average number of retransmissions, which increases as the number of devices increases, but interference cancellation can reduce the number of retransmissions, even for good-channel devices. 

### 5.2. When a Good-Channel Device Performs Random Backoff

In the simulations in the previous subsection, good-channel devices do not perform random backoff in units of the superframe. This subsection considers a case where good-channel devices perform random backoff in units of the superframe in addition to randomly selecting a slot position in the superframe. By allowing good-channel devices to backoff first, and then, check the response, interference cancellation for good-channel devices can be performed, and the collision probability of good-channel devices can also be reduced. The contention window size for good-channel devices is set to four superframes. Assuming that a good-channel device cannot transmit immediately in the next superframe when retransmitting, it performs a random backoff from one to four superframes based on a random number selected from the contention window. The other simulation parameters are the same as in the previous simulations. 

Figure 15 shows the result of the same setup as in Figure 9, except that good-channel devices perform random backoff. The figure shows the collision probability of a bad-channel device as a function of the number of devices without interference cancellation, as the packet length of a bad-channel device increases from 1 to 16. While the slots within the superframe are randomly selected in Figure 9, a good-channel device additionally performs random backoff in Figure 15, but there is not much difference in the collision probability from Figure 9. As the packet length increases, the collision probability rises sharply, and the number of retransmissions increases, which can shorten the lifetime of the bad-channel device. 

Figure 16 shows the result of the same setup as in Figure 10, except that good-channel devices perform random backoff. A packet from a bad-channel device occupies 16 slots, and delay 1 in the figure means a delay of 16 slots. In the case of Figure 10, the delay number refers to the number of previously transmitted packets considered for interference cancellation. However, in Figure 16, random backoff is used for good-channel devices, and a much smaller number of previously transmitted packets is considered for interference cancellation at the same delay number. Therefore, the performance improvement is not significant for small delay values. If a good-channel device has a large retransmission interval due to random backoff, the response delay of a bad-channel device should be very large to ensure sufficient interference cancellation. If the response delay of a bad-channel device is limited to a small value, it is necessary to reduce the random backoff of a good-channel device so that a sufficient number of retransmissions can occur during the response delay. 

Figure 17 shows the result of the same setup as in Figure 11, except that good-channel devices also perform random backoff. Compared to Figure 16, a bad-channel device uses Random Backoff First instead of Delayed Response. A bad-channel device performs Random Backoff First and checks the response to determine whether to retransmit. Compared to Figure 11, which uses a fixed interval between retransmitted packets for good-channel devices, the random backoff can increase the interval between retransmitted packets for good-channel devices. If a bad-channel device selects a small value in the contention window, good performance cannot be achieved because sufficient retransmissions may not be performed by good-channel devices. If a bad-channel device uses Random Backoff First instead of Delayed Response, the contention window for good-channel devices must be much smaller than that for the bad-channel device to ensure sufficient interference cancellation. It may also be useful to use both Delayed Response and Random Backoff First together so that sufficient interference cancellation can be performed even if a small value is selected from the contention window of the bad-channel device. 

Figure 18 and Figure 19 show the result of the same setup as in Figure 13 and Figure 14, except that good-channel devices perform random backoff. In the simulation, all devices have good channel conditions, and each packet occupies one slot. In Figure 18, by checking the response after random backoff, some interference can be eliminated if a random number chosen from the contention window is not small, and the performance improvement due to interference cancellation is significant compared to Figure 13. 

Figure 19 shows the average number of retransmissions. Even when interference cancellation is not performed, the number of retransmissions can be reduced to some extent by random backoff compared to Figure 14. When interference cancellation is performed, the number of retransmissions can be significantly reduced compared to Figure 14. Performing interference cancellation can help not only bad-channel devices but also good-channel devices, especially when random backoff is performed for good-channel devices. As the contention window for good-channel devices increases, the performance of good-channel devices can be improved, but there is the drawback of requiring a larger response delay for a bad-channel device. Depending on how much response delay a bad-channel device can tolerate, it may be necessary to determine the contention window for good-channel devices. 

## 6. Conclusions

This paper proposes methods for IoT devices to reduce the collision probability of bad-channel devices while using shared resources, instead of separating resources between groups depending on the channel conditions. Two versions of the scheme are proposed: in the first method, the response of a bad-channel device is delayed, while in the second method, a bad-channel device first performs random backoff before checking the response. Before transmitting the response to a bad-channel device, the base station eliminates interference from other devices using the retransmitted packets from good-channel devices. The proposed methods can reduce the collision probability of bad-channel devices and increase their lifetime without increasing the transmission power or collision probability of good-channel devices. When properly performed, the interference cancellation can reduce the collision probability of good-channel devices in addition to bad-channel devices. 

In the future, further analysis and experimentation is needed on how to deal with a mixture of devices with various channel conditions and when there are many bad-channel devices. Furthermore, the impact of imperfect interference cancellation needs to be analyzed, and theoretical and experimental comparisons with other interference cancellation techniques, such as NOMA and coded random access, are needed. 

## Figures and Tables

**Figure 1 sensors-23-09556-f001:**
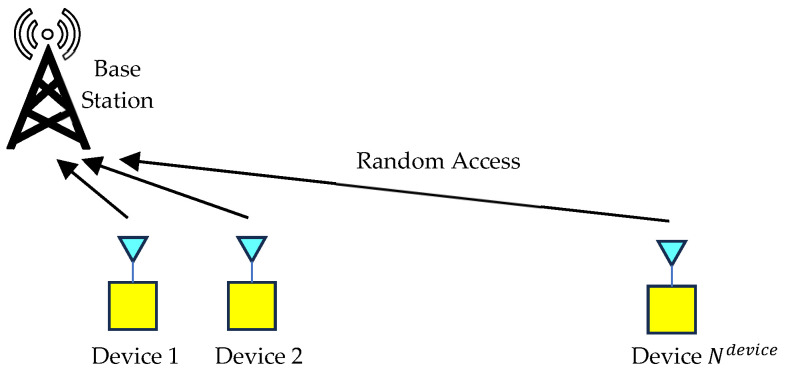
IoT system.

**Figure 2 sensors-23-09556-f002:**
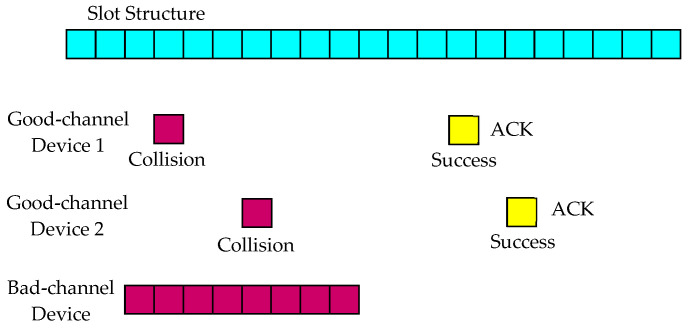
Collision when the number of repetitions is large.

**Figure 3 sensors-23-09556-f003:**
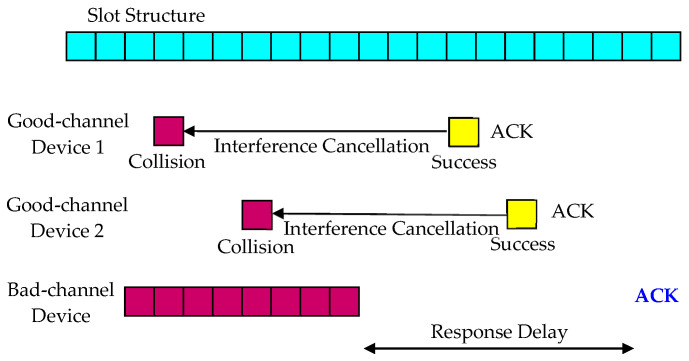
Interference cancellation in the Delayed Response method.

**Figure 4 sensors-23-09556-f004:**
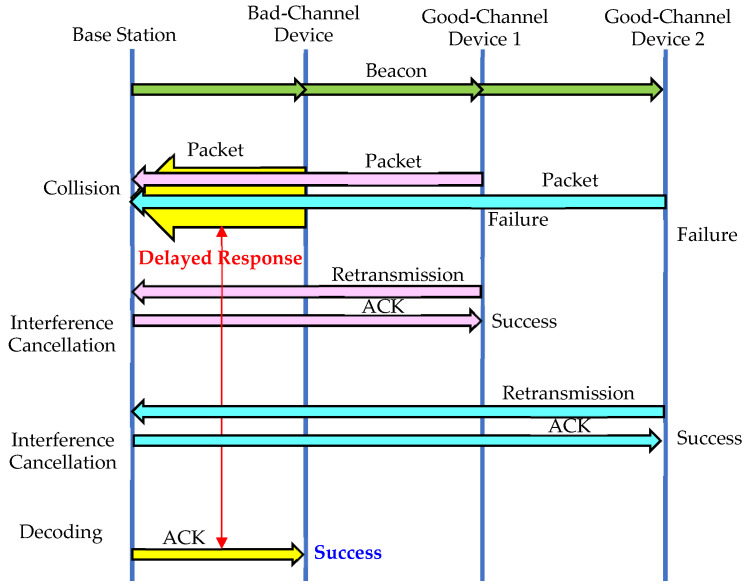
Operations for the Delayed Response method.

**Figure 5 sensors-23-09556-f005:**
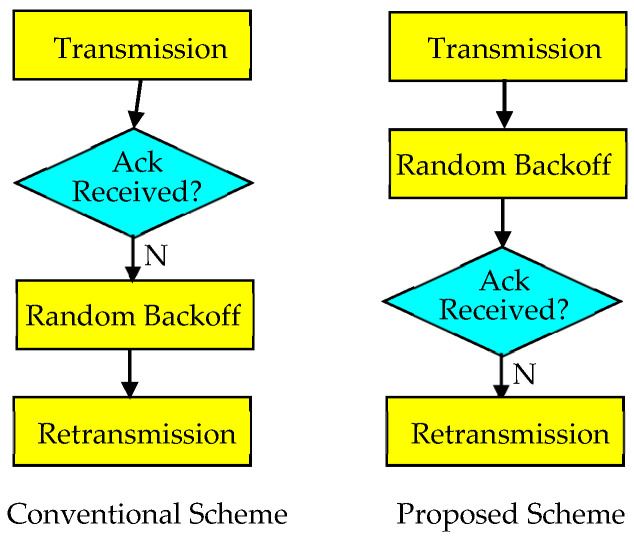
Comparison between conventional and Random Backoff First schemes.

**Figure 6 sensors-23-09556-f006:**
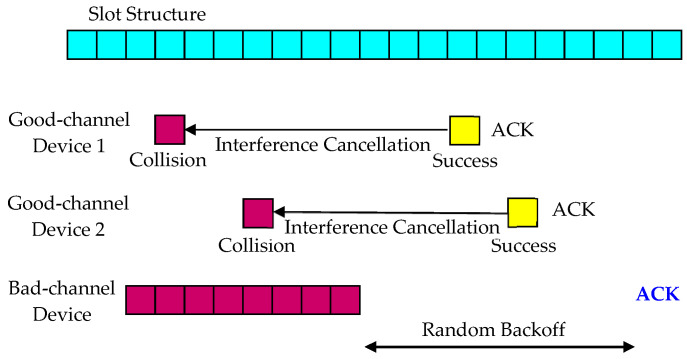
Receiving a response after a random backoff.

**Figure 7 sensors-23-09556-f007:**
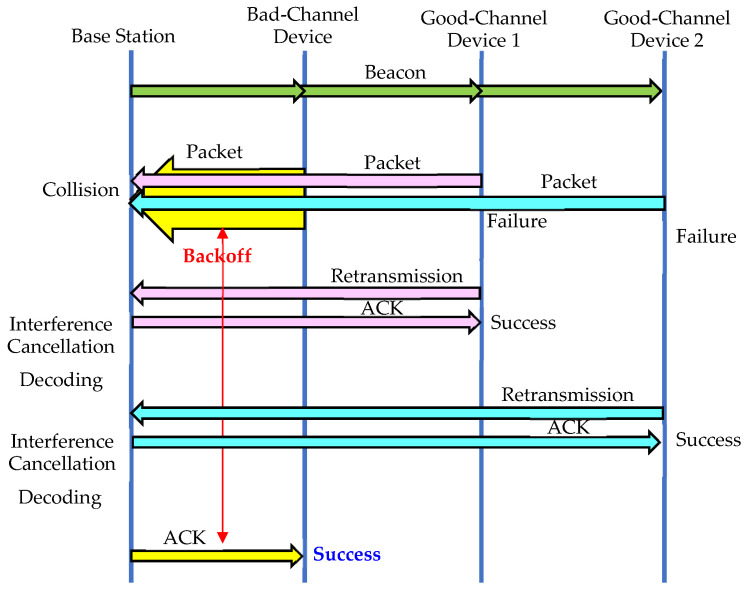
When a large value is selected in the contention window.

**Figure 8 sensors-23-09556-f008:**
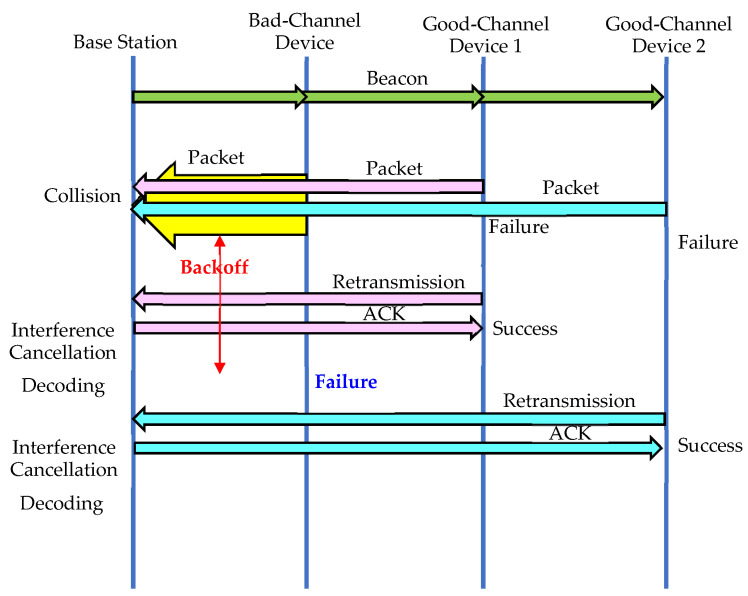
When a small value is selected in the contention window.

**Figure 9 sensors-23-09556-f009:**
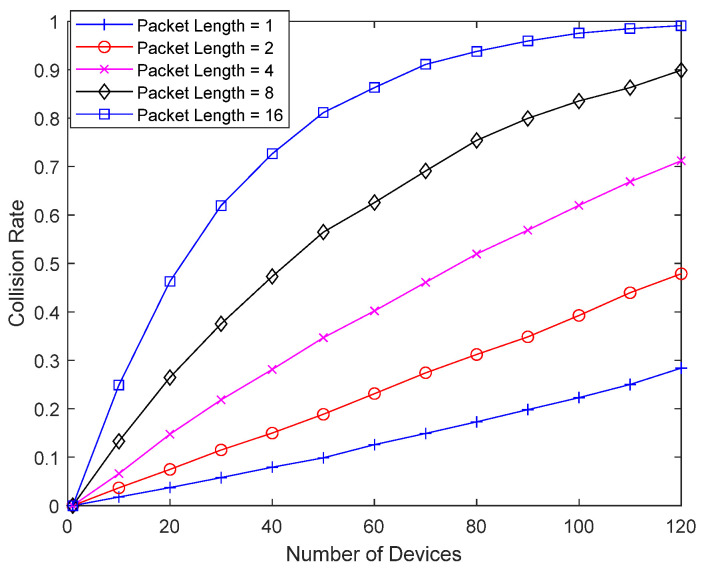
Collision probability according to packet length.

**Figure 10 sensors-23-09556-f010:**
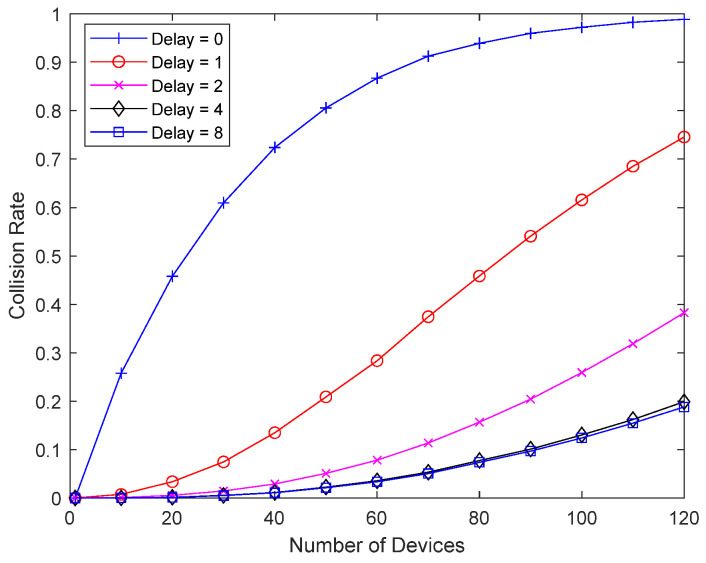
Collision probability according to response delay.

**Figure 11 sensors-23-09556-f011:**
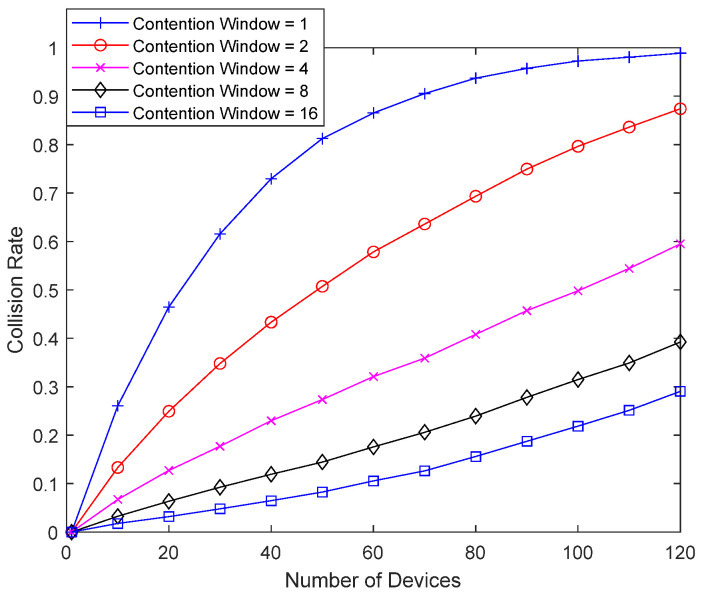
Collision probability according to contention window size.

**Figure 12 sensors-23-09556-f012:**
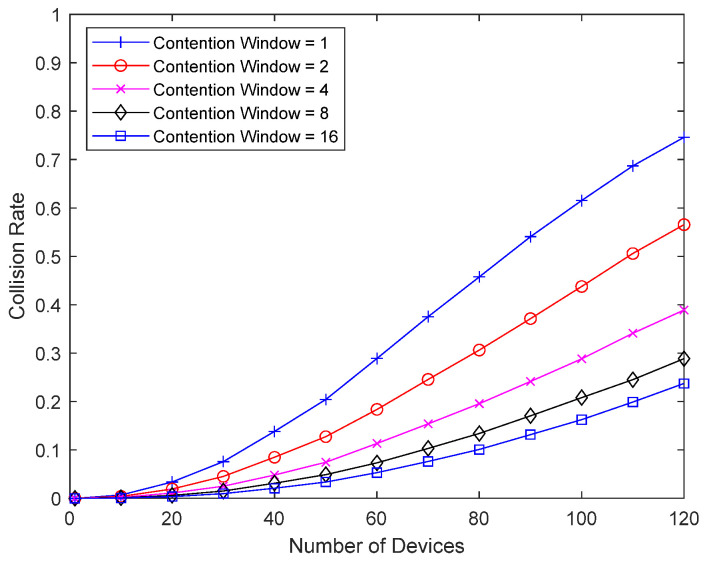
Collision probability according to contention window size when Delayed Response is also applied.

**Figure 13 sensors-23-09556-f013:**
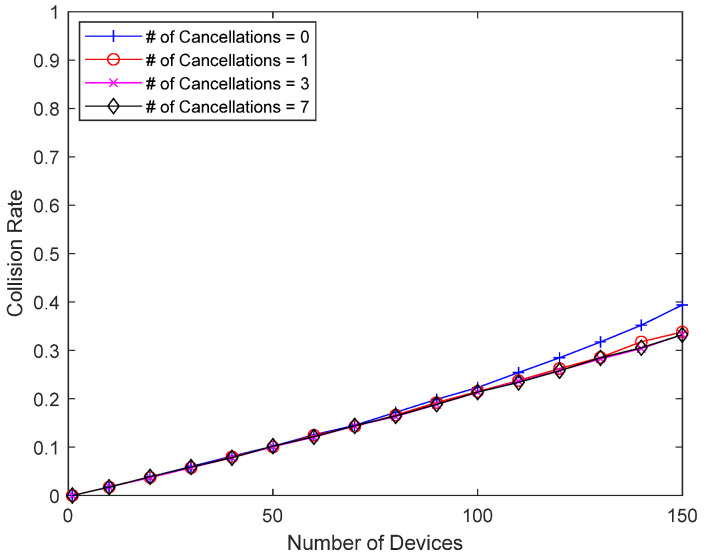
Collision probability of good-channel devices according to the number of packets undergoing interference cancellation.

**Figure 14 sensors-23-09556-f014:**
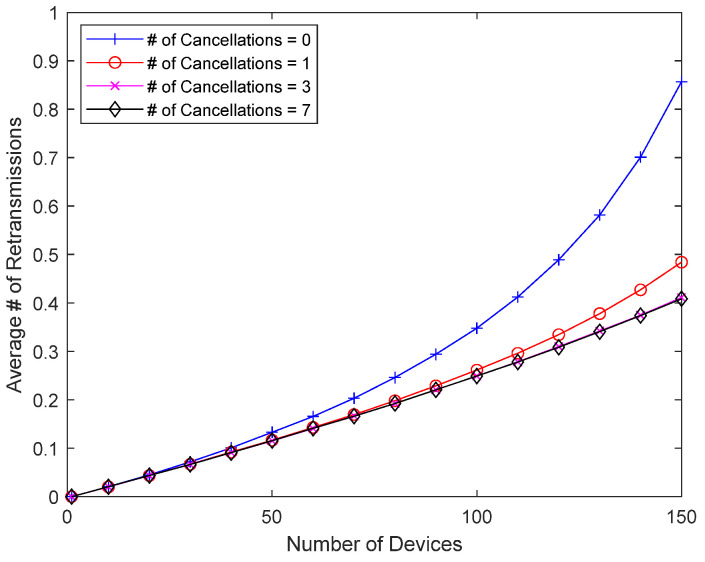
Number of retransmissions according to the number of packets undergoing interference cancellation.

**Figure 15 sensors-23-09556-f015:**
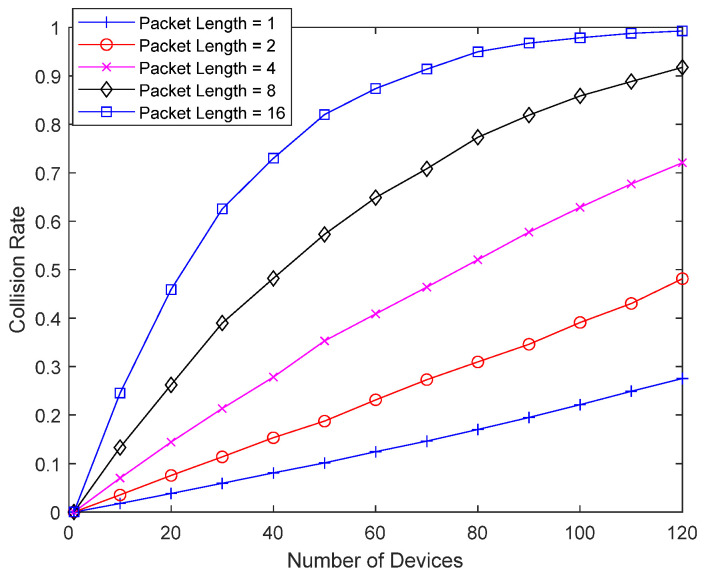
Collision probability according to packet length.

**Figure 16 sensors-23-09556-f016:**
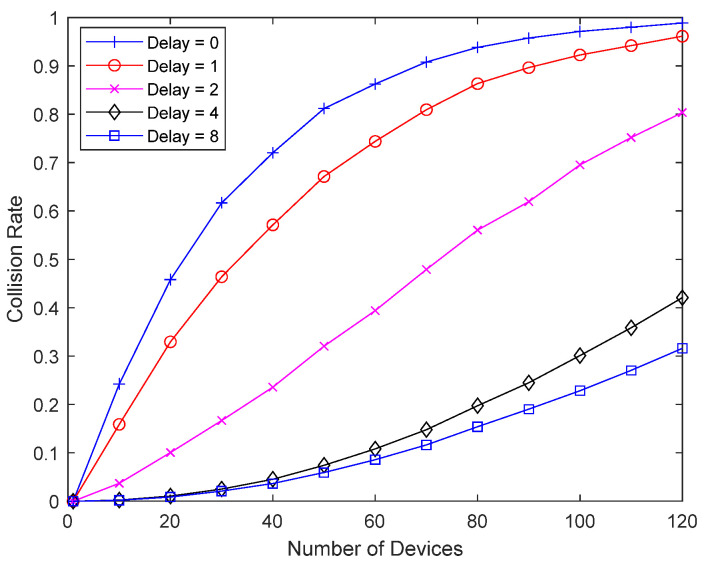
Collision probability according to response delay.

**Figure 17 sensors-23-09556-f017:**
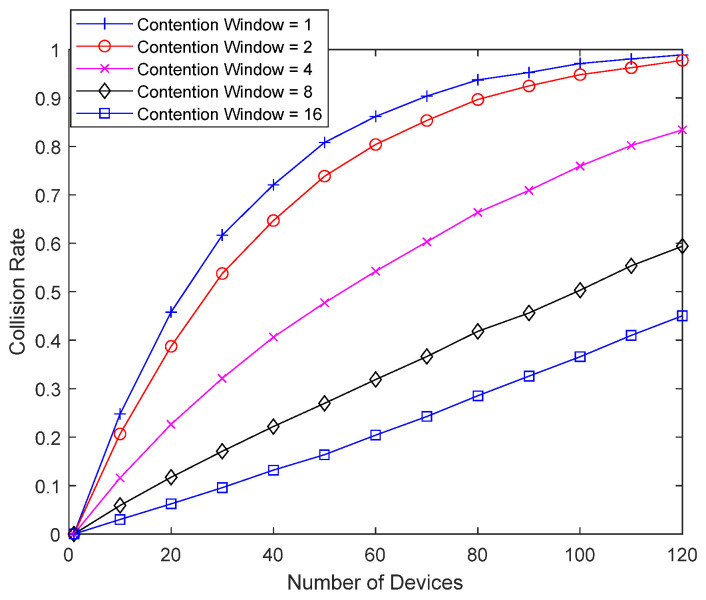
Collision probability according to contention window.

**Figure 18 sensors-23-09556-f018:**
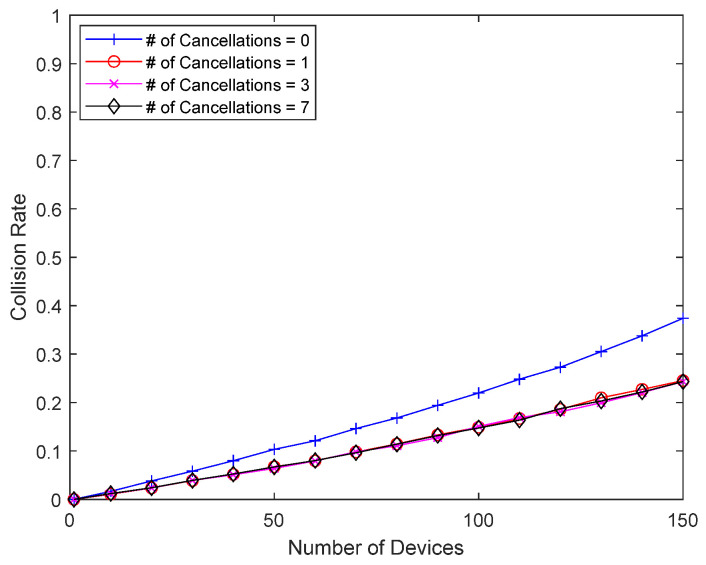
Collision probability of good-channel devices according to the number of packets undergoing interference cancellation.

**Figure 19 sensors-23-09556-f019:**
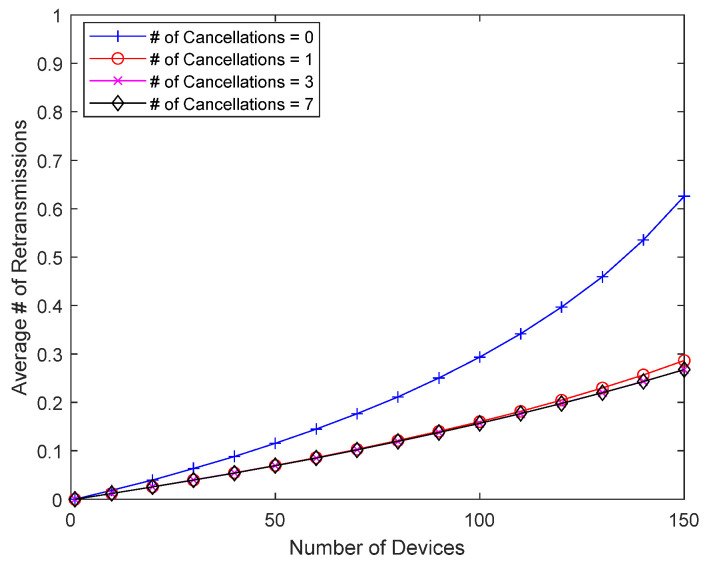
Number of retransmissions according to the number of packets undergoing interference cancellation.

**Table 1 sensors-23-09556-t001:** Differences between the Delayed Response and Random Backoff First methods.

Delayed Response	Random Backoff First
Delays response	Reverses the order of response checking and random backoff
Increases transmission latency	Does not increase transmission latency
Attempts to decode once just before the response time	Attempts to decode multiple times each time interference cancellation is performed
Performs sufficient interference cancellation if the delay value is large	May perform insufficient interference cancellation if a small backoff value is selected

**Table 2 sensors-23-09556-t002:** Simulation parameters.

Simulation Parameters	Values
Number of devices performing random access ($N^{device}$)	1~120 for Figure 9, Figure 10, Figure 11 and Figure 12; 1~150 for Figure 13 and Figure 14; 1~120 for Figure 15, Figure 16 and Figure 17; 1~150 for Figure 18 and Figure 19
Virtual frame size (M)	8
Packet length of a good-channel device	1
Transmission period	One packet per 512 slots
Packet length of a bad-channel device	1, 2, 4, 8, 16 for Figure 9; 16 for Figure 10, Figure 11 and Figure 12; 1 (i.e., no bad-channel device) for Figure 13 and Figure 14; 1, 2, 4, 8, 16 for Figure 15; 16 for Figure 16 and Figure 17; 1 (i.e., no bad-channel device) for Figure 18 and Figure 19
Eliminating interference caused by a good-channel device	0 packets for Figure 9; Up to 7 previously transmitted packets for Figure 10, Figure 11, Figure 12, Figure 13 and Figure 14; 0 packets for Figure 15; Up to 7 previously transmitted packets for Figure 16, Figure 17, Figure 18 and Figure 19
Random access schemes for a bad-channel device	Conventional for Figure 9; Delayed Response for Figure 10; Random Backoff First for Figure 11; Delayed Response + Random Backoff First for Figure 12; No bad-channel device for Figure 13 and Figure 14; Conventional for Figure 15; Delayed Response for Figure 16; Random Backoff First for Figure 17; No bad-channel device for Figure 18 and Figure 19
Contention window size of a good-channel device	1 for Figure 9, Figure 10, Figure 11, Figure 12, Figure 13 and Figure 14; 4 for Figure 15, Figure 16, Figure 17, Figure 18 and Figure 19
Contention window size of a bad-channel device	1, 2, 4, 8, 16 for Figure 11, Figure 12 and Figure 17

## Data Availability

Data are contained within the article.

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
