# Peer review of "Delayed Response and Random Backoff First for Low-Power Random Access of IoT Devices with Poor Channel Conditions"

_sensors, 2023, doi:10.3390/s23239556_

Round 1
Reviewer 1 Report
Comments and Suggestions for Authors
The work presents two methods to reduce collision probability of bad-channel devices for IoT applications. Both methods are related to the time the receiver sends the response to the transmitting entity. The work is well written and there are plenty results to back up their results.
Though there are some issues that should be resolved in order to enhance the readership of this work:
1)line 314. Why the good-channel devices should delay their transmission one superframe time?please elaborate
2)line 323? please provide the parameters for the simulations in a table form as there are missing variables like message distribution.
3)Please provice an algorithmic diagram (flowchart or statechart) for the proposed algorithms.
4)As the delay is mentioned in the document and the algorithm adds delay , the authors should perform simulations to find the additional delay their algo inserts for one case for each algorith at least..
5)line 347 cacellation..
Reviewer 2 Report
Comments and Suggestions for Authors
In this paper, two methods to reduce the collision probability of bad-channel devices while allowing IoT devices
to use shared resources. The research undertaken is the need for IoT, especially LPWANs where the battery is one of the major constraints. The research shows lower collision probabilities even in bad channel conditions.
- Second last statement in the abstract: "The second method is to first perform a random backoff, and then check the response to decide whether to retransmit, instead of checking the response and performing a random backoff if no acknowledgment packet is received" This Needs to be rewritten for clarity and to increase readability.
- Recently reported methods related to energy efficient transmission based on multi-armed bandits algorithm for LoRa performance enhancement and such need to be discussed.
- the flowcharts given in the papers are very primitive. Details of both methods depicting end-to-end transmission/reception with a closed loop can be presented.
- Equations and parameters need to be rechecked.
- Why do NACK is needed? A wait timer can be added.
- how the proposed methods respond wrt packet delivery rate and energy consumption. Such results would increase interest among with readers.
- Time/Computational complexity needs to be included as it is concerned with IoT/low-power applications
- All simulation parameters and setup need to be explained well.
Comments on the Quality of English Language
English is fairly good.
Reviewer 3 Report
Comments and Suggestions for Authors
This paper proposed two methods to avoid collision in IoT networks.
1- Figures 1 should be modified it doesn't represent the proposed model.
2- The two proposed methods should be compared to show the difference between them.
3- The authors discussed interference avoidance, but no figure represents this issue.
4- The proposed model should be compared with proposed model in the literature.
Round 2
Reviewer 1 Report
Comments and Suggestions for Authors
The authors have done an extensive review, covering almost all the issues and enhancing the readership to a great extent. Numerous additions were made, providing enough details about their work.
The only point I would like to see some additions is the simulation parameter table, as there is no information about the data arrival rate (data distribution used to feed the system: was it a Pareto or a normal distribution? Please add this to the parameter table.)
Author Response
Your detailed and valuable comments on the revision of the manuscript are greatly appreciated.
The simulation parameter table has been updated.
Reviewer 2 Report
Comments and Suggestions for Authors
-
Comments on the Quality of English Language-
Author Response
Your detailed and valuable comments on the revision of the manuscript are greatly appreciated.
The manuscript was proofread and edited by the professional English editors.

Reviewer 3 Report
Comments and Suggestions for Authors
The authors have addressed the required comments.
Author Response
Your detailed and valuable comments on the revision of the manuscript are greatly appreciated.